# Physical Activity, Anxiety, Depression, and Body Image in Trans Individuals: An Exploratory Study

**DOI:** 10.3390/healthcare12101008

**Published:** 2024-05-14

**Authors:** Joana Oliveira, Diogo Monteiro, Miguel Jacinto, Rui Matos, Nuno Amaro, Filipe Rodrigues, Raúl Antunes

**Affiliations:** 1ESECS—Polytechnic of Leiria, 2411-901 Leiria, Portugal; 1220237@my.ipleiria.pt (J.O.); miguel.s.jacinto@ipleiria.pt (M.J.); rui.matos@ipleiria.pt (R.M.); nuni.amaro@ipleiria.pt (N.A.); filipe.rodrigues@ipleiria.pt (F.R.); raul.antunes@ipleiria.pt (R.A.); 2Research Center in Sport Sciences, Health Sciences and Human Development (CIDESD), 5001-801 Vila Real, Portugal

**Keywords:** physical activity, body image, anxiety, depression, transgender

## Abstract

Physical activity (PA), mental health, and body image are some important health topics in the transgender population that have been recently discussed and appear to play a crucial role in the quality of life of the trans population. This study aims to elucidate the complex interplay of these variables and their implications for the well-being of trans individuals. Methods: In a cross-sectional study, 75 Portuguese transgender individuals (M = 23.68; SD ± 6.59) were recruited to participate in this study. The participants completed three questionnaires related to the assessment of physical activity (IPAQ), depressive and anxious symptomatology (HADS), and satisfaction with body image (BISQp). Results: Trans individuals showed a total energy expenditure of 3316.40 metabolic equivalent tasks (METS), had a moderate level of anxiety symptomatology, and low levels of satisfaction with body image. Satisfaction with body image was negatively associated with anxiety (r = −0.441, *p* < 0.01) and depression symptomatology (r = −0.600, *p* < 0.01). Conclusions: The implementation of inclusive programs that promote body acceptance and coping strategies, particularly within the context of physical exercise, may help alleviate distress related to body image dissatisfaction while also addressing underlying anxiety and depression symptoms.

## 1. Introduction

The term transgender or trans describes individuals whose gender identity diverges from the sex that was assigned to them at birth [1,2,3]. In 2018, the ICD-11 presented a significant change in the approach to transgender identities and gender diversity. In contrast to previous versions, the ICD-11 no longer uses terms like “transsexualism” and “gender identity disorder”. Instead, it introduces the concept of “gender incongruence” to describe situations where an individual’s gender identity does not align with the sex assigned at birth. Additionally, the ICD-11 also acknowledges and encompasses a diverse range of gender identities beyond the binary classification of male and female [4]. In recent years, there has been increasing recognition of non-binary identities within the transgender community. A survey conducted in the United Kingdom revealed that approximately 52% of transgender respondents identified as non-binary [5]. This highlights the diversity within transgender experiences, which extends beyond traditional binary notions of gender to encompass a spectrum of identities including non-binary and genderqueer individuals. This diversity challenges conventional understandings of gender as a strictly binary concept and underscores the importance of acknowledging and affirming non-binary and genderqueer identities within discussions on transgender issues [6,7].

Trans populations often experience discrimination [8], difficulties in accessing healthcare [9], and report numerous obstacles to engaging in physical activity (PA) [10], which explains the high prevalence of mental health problems, namely anxiety and depression [8,11,12,13], greater dissatisfaction with body image related to gender dissociation and dissatisfaction with body shape [14,15], and lower levels of PA [16]. In fact, PA, mental health, and body image are some important health topics in the transgender population that have been recently discussed and appear to play a crucial role in the quality of life of this population [12,17,18,19].

The recent systematic review by Schweizer and colleagues [20] suggests that transgender individuals have PA levels below recommendations and lower levels compared to cisgender individuals. This can be explained by the existence of significant barriers that hinder more frequent engagement in PA in this population, namely body dissatisfaction and discomfort, discrimination, anxiety about others’ reactions, and lack of safe and comfortable spaces [10]. Socio-cultural acceptance, as discussed by Monaco and Corbisiero [21], along with the existence of progressive legal policies that prioritize transgender healthcare rights [22], or conversely, the presence of discriminatory policies that create barriers to healthcare access [23] are also important aspects that can contribute to disparities in PA levels within this population. This is particularly concerning given the significant relationship established in the literature between PA and mental health.

The regular role of PA in individuals’ quality of life has been studied over time, and its benefits for psychological well-being are evident [24]. Specially concerning mental health, regular PA is associated with improved mental well-being and the prevention of anxiety and depression disorders [25,26]. In contrast, sedentary behavior and physical inactivity have been reported as risk factors for depression and anxiety [27]. Body image has been identified as a moderator of this relationship, indicating that the association between physical activity and anxiety varies depending on individuals’ perceived levels of body image. Specifically, individuals with a heightened awareness of body image tend to exhibit a stronger link between physical activity and anxiety levels [28].

These data are particularly important for populations with a high prevalence of depressive and anxious symptoms, such as the transgender community [12,19]. Additionally, in cisgender samples, higher levels of anxiety and depression appear to be associated with greater dissatisfaction with body image [29,30], which can in turn be positively influenced by PA [31]. In fact, in the study by Bassett-Gunter and colleagues [32], the results suggest a positive relationship between PA and body image, mainly among men.

Body image, intricately linked with societal norms and expectations, appears to play a pivotal role in the mental health of transgender individuals. In the study conducted by Röder and colleagues [33], the results revealed that less satisfaction with body image significantly predicted lower health-related quality of life (HRQoL) outcomes. Similar results were also found in other studies, where the findings clearly demonstrate significant correlations between self-esteem, depression, and body image in individuals with gender dysphoria [34].

The relationship between body image and anxiety and depression in transgender individuals is bidirectional and complex. Previous research indicates that poor body image can contribute to heightened negative emotions regarding specific gender characteristics and body parts [35]. In fact, the parts of the body that caused the most dissatisfaction among trans individuals were the beard, body hair, skin, Adam’s apple, chest/breast, stomach, waist, hips, and bottom [15]. Additionally, the findings revealed that anxiety (both regarding self-image and social situations) and perfectionism (both criticism and control) were identified under body dissatisfaction, suggesting that symptoms of anxiety can exacerbate body image concerns, leading to a vicious cycle of negative self-perception and emotional distress [14].

### Current Study

The existing literature has demonstrated negative correlations between anxiety and depression and satisfaction with body image and PA [28], as well as positive correlations between satisfaction with body image and PA [29]. However, despite the growing body of literature on transgender health, there exists a notable gap in a thorough and comprehensive characterization of transgender individuals, particularly concerning their satisfaction with body image, depressive and anxious symptoms, and engagement in PA. This study seeks to address this gap by conducting an exploratory analysis to elucidate the complex interplay of these factors and their implications for the well-being of trans individuals, based on a detailed characterization of the sample. The insights gained from this research could lead to rigorous and scientific knowledge about the extent and multifaceted nature of mental health and body image concerns within this community and contribute to the destigmatization of transgender identities and the development of tailored interventions to promote mental well-being and quality of life.

In Portugal, legislation has been implemented granting the right to self-determination of gender identity and gender expression, along with protections for individuals’ sexual characteristics. Moreover, legal recognition of gender identity is provided for transgender youth between the ages of 16 and 18 [36]. However, the Portuguese society continues to face challenges in accepting and integrating transgender individuals. Research has highlighted that discrimination against transgender individuals by healthcare professionals is perceived by a significant portion of those seeking care and access to gender-affirming healthcare services, through the Portuguese National Health Service, may be limited and subject to long waiting periods [37]. Additionally, transgender individuals in Portugal encounter extensive societal challenges and discrimination, making them one of the most marginalized and excluded populations in the country [38]. Specific legislation addressing the participation of transgender individuals in sports is currently lacking [36].

The primary aim of this study is to identify levels of PA, anxious and depressive symptomatology, and satisfaction with body image, and to examine the relationship between these three variables. We hypothesize the following: (a) transgender individuals will report high levels of anxious and depressive symptoms, with low levels of PA and body image satisfaction; (b1) a negative and significant correlation will emerge between PA and anxious and depressive symptomatology, as the literature has reported in different samples (e.g., [39,40,41]); (b2) a negative and significant correlation will emerge between anxious and depressive symptomatology and satisfaction with body image, supported by previous studies (e.g., [29,42,43]).

## 2. Materials and Methods

### 2.1. Study Design and Participants

This cross-sectional study conducted in Portugal aims to examine the relationship between physical activity, mental health, and body image in transgender people. The sample consisted of a total of 75 (*n* = 75) Portuguese transgender individuals (23.68 ± 6.59 years of age), of which 62 (82.7%) were trans men and 13 (17.3%) were trans women. A majority of 54 participants (72%) were in the process of transitioning. The mean age at transition initiation was 21.37 ± 6.44 years old. Participants reported recognizing their gender identity misalignment at a mean age of 16.09 ± 7.05 years old. Furthermore, they began openly and consciously expressing their gender identity on average at 18.39 ± 7.45 years old.

All participants were fully informed about the nature of the study and the procedures for data recording. Before completing the survey, informed consent was obtained from each participant individually. All participants participated in this study voluntarily and confidentiality and anonymity were guaranteed. They were also informed that they could withdraw from the study at any time. The sample size for regression analysis was calculated using G*Power 3.1.9.7 [44], based on the following input parameters: effect size (f^2^ = 0.3), α = 0.05, and statistical power = 0.95. The minimum required sample size was determined to be 55 subjects, which was adhered to in the current study.

### 2.2. Procedure: Data Collection

Prior to data collection, ethical approval was obtained from the Ethics and Scientific Committee of the Polytechnic of Leiria, under reference number CE/IPLEIRIA/47/2023. The present study was conducted in accordance with the Declaration of Helsinki [45].

For data collection, a Google form was used as the survey platform. The survey consisted of sociodemographic questions and questionnaires validated for the Portuguese population that assessed 3 domains: levels of PA, anxiety and depressive symptoms, and satisfaction with body image. Respondents took an average time of 10 min to complete the survey. Additionally, Portuguese LGBTQ+ associations and institutions were contacted, and meetings were conducted. During these meetings, the researchers elucidated the study’s objectives and addressed any inquiries. Subsequently, support for conducting the study was solicited from these associations and institutions to publicize the study and recruit participants, through their networking and digital platforms.

### 2.3. Instruments

#### 2.3.1. Physical Activity

To assess participants’ levels of physical activity, the short form of the International Physical Activity Questionnaire (IPAQ), validated for 12 countries, including Portugal, was used [46]. The questionnaire comprises a total of nine questions related to activities performed in the last seven days prior to questionnaire application [46]. The questions assess principles of physical activity, such as walking, moderate-intensity activities, and vigorous activities, including their frequency and duration. Coding involves estimating energy expenditure based on levels of physical activity. Specifically, the obtained data are converted into MET (metabolic equivalent of task) minutes per week. To calculate MET minutes per week, one multiplies the MET values (walking = 3.3; moderate-intensity physical activity = 4; vigorous physical activity = 8) by the number of minutes of activity performed each day over the last 7 days.

#### 2.3.2. Depressive and Anxious Symptomatology

The Portuguese version of the Hospital Anxiety and Depression Scale (HADS) [47] was used to assess depressive and anxious symptomatology. This scale consists of 14 items, with 7 items dedicated to measuring anxiety symptoms (e.g., “I feel tense or nervous”) and the other 7 to measuring depression symptoms (e.g., “I have lost interest in my physical appearance”). Each item is scored on a Likert scale ranging from 0 to 3, with higher scores indicating greater symptom severity. Subsequently, the scale yields two dimensions: depression symptomatology and anxiety symptomatology. The total depression score is calculated as the sum of questions 2, 4, 6, 8, 10, 12, and 14, while the total anxiety score is the sum of questions 1, 3, 5, 7, 9, 11, and 13. The dimensions are individually categorized as follows: ≤7 points—absence of symptomatology; 8–10 points—mild symptomatology; 11–14 points—moderate symptomatology; 15–21 points—severe symptomatology. Internal consistency reliability in this study proved to be good (anxiety α = 0.83; depression α = 0.81).

#### 2.3.3. Satisfaction with Body Image

The Body Image Satisfaction Questionnaire Portuguese Version [48] was employed to assess participants’ satisfaction with various body features such as nose, hair, and shoulders, as well as their overall satisfaction with body image encompassing vitality, body shape, and physical appearance. This questionnaire consists of 23 items focusing on facial features (e.g., teeth, hair, eyes, nose), body parts (e.g., glutes, arms, chest), and overall appearance factors (e.g., physical fitness, height, vitality, body shape).

The questions are presented on a Likert scale with a five-point range from 1 (“I don’t like and would like to be different”) to 5 (“I consider myself favored”). Thus, body image satisfaction is derived from the average of the 23 items, and the higher this average, the greater the individual’s satisfaction with their body image. Internal consistency reliability in this study proved to be good (face α = 0.85; upper trunk α = 0.76; lower trunk α = 0.87; legs and glutes α = 0.88; body appearance α = 0.83).

### 2.4. Statistical Analysis

Data analyses were performed using the IBM SPSS software for Windows (Version 29.0, IBM Corp, Armonk, NY, USA). Counts (and proportions), means, standard deviations (SD), 95% confidence interval (95% CI), and medians (interquartile range, IQR) were calculated to describe both categorical and continuous variables for the total sample.

Missing values and outliers were examined within the dataset. Pearson’s correlation coefficients were computed to assess the relationships among the variables of interest. The interpretation of correlation strength followed guidelines proposed by Hopkins and colleagues [49], categorizing correlations as trivial (r < 0.1), small (0.1 < r < 0.3), moderate (0.3 < r < 0.5), large (0.5 < r < 0.7), very large (0.7 < r < 0.9), and nearly perfect (r ≥ 0.9). The multivariate regression analysis was fitted to explain the anxious and depressive symptomatology based on predictive factors, namely satisfaction with body image relative to the face, upper trunk, lower trunk, legs and glutes, and body appearance. For all tests, the level of significance was set at *p* < 0.05 to reject the null hypotheses Ho [50].

## 3. Results

### 3.1. Descriptive Statistics

Data inspection did not reveal missing values and outliers. The sample characteristics are presented in Table 1. Regarding PA, the highest values are found in moderate-intensity PA (1262.67 METS), followed by vigorous-intensity PA (1195.73 METS) and light-intensity PA (858.73 METS). Trans individuals showed a total energy expenditure of 3316.40 METS. Descriptive statistics also indicated higher values for anxiety symptoms (11.67) compared to depression symptoms (7.41). Moreover, the sample had scores above the midpoint especially for anxious symptoms, indicating a moderate level of symptomatology. Finally, regarding satisfaction with body image, the overall mean score was low (SBI—Global = 2.82), with the lower trunk (SBI—Lower Trunk = 2.18) and upper trunk (SBI—Upper Trunk = 2.51) components having the lowest scores. Conversely, satisfaction with body image related to the face had the highest score (SBI—Face = 3.51).

### 3.2. Bivariate Correlations

As expected, several significant bivariate correlations emerged, as can be seen in Table 2. All the components of satisfaction with body image (face, upper and lower trunk, legs and glutes, body appearance, and global satisfaction with body image) were negatively associated with anxiety and depression symptomatology. Regarding the dimension of PA, only one significant correlation emerged, specifically concerning vigorous PA, which was positively associated with satisfaction with body image related to the lower trunk (r = 0.231, *p* < 0.05). The other components of PA were negatively associated with anxious and depressive symptoms; however, these results were not statistically significant.

### 3.3. Multiple Regressions

Model 1 (Table 3) with all the predictors together (SBI—Face; SBI—Upper Trunk; SBI—Lower Trunk; SBI—Legs and Glutes; SBI—Body Appearance) to explain the variable anxious symptomatology, significantly fitted the data (F = 4.39; *p* = 0.002), jointly explaining 24% of the variance of anxious symptomatology. Additionally, the regression coefficient results of the different body image subscales for depression were negative and significant (*p* < 0.05).

Model 2 (Table 4) with all the predictors together (SBI—Face; SBI—Upper Trunk; SBI—Lower Trunk; SBI—Legs and Glutes; SBI—Body Appearance) to explain the variable anxious symptomatology, significantly fitted the data (F = 4.39; *p* = 0.002), jointly explaining 24% of the variance of anxious symptomatology. Additionally, the regression coefficient results of the different body image subscales for depression were negative and significant (*p* < 0.05).

## 4. Discussion

There is a lack of exploratory studies providing a detailed characterization of the transgender population concerning their body image, levels of anxiety and depression, and engagement in PA, while simultaneously exploring the relationship between these health topics. To fill this gap in the literature, this study aims to identify levels of PA, anxious and depressive symptomatology, and satisfaction with body image in trans individuals and to examine the interplay between these variables.

### 4.1. Physical Activity

Regarding PA levels, the current evidence-based recommendations suggest that for substantial health benefits, adult individuals should complete at least 150 min of moderate-intensity PA or 75 min of vigorous-intensity PA or an equivalent combination of moderate and vigorous-intensity PA achieving at least 600 MET min/week [51]. However, major health gains occur at a total activity level of 3000–4000 MET min/week [52]. In the current study, the levels reported by participants range from 2502 to 4130 MET min/week, with highest values in moderate-intensity PA, which suggests significant PA practice in this population with benefits. Similar results related to the levels of PA in transgender samples were also found in another studies. In the recent work by Ceolin and colleagues [53], the transgender individuals reported values of PA between 525.00 and 2772.00 METS, with no statistically relevant differences between the trans and cis sample. Data from the Minnesota Transgender Youth Survey also revealed that the majority of individuals (50.2%) are physically active for at least 60 min on 3 or more days per week [54]. These are positive and quite surprising results, considering that previous evidence has consistently identified lower PA levels in trans individuals and reinforced the significant difference between the trans and cis population in this regard. Smalley and colleagues [55] found that only 36.9% of trans men and 24.3% of trans women met the lowered recommendation of 20 min per day, three days per week of PA. Similar, Cunningham and colleagues [56] reported no PA in the past month for 43.3% of trans women and 31% of trans men.

The inconsistency of the results of this study with most of the literature could be explained by the bivalent way in which physical activity can be perceived by the trans population. In the recent scoping review by Schweizer and colleagues [20], the results suggest that trans people engage in low levels of PA or in compulsive exercise, which reflect the complexity of this topic and the diverse experiences within the transgender community. It is essential to recognize that the relationship between trans individuals and PA is not uniform. Some may find joy and empowerment in PA, while others may struggle due to various barriers and challenges, which may contribute to increased health risks in this population.

### 4.2. Anxiety and Depression

In the present study, anxiety values were higher than depression. There was a moderate anxious symptomatology and a normal depressive symptomatology found in the sample, according to the reference values of Pais-Ribeiro and colleagues [47]. This has been observed in previous studies. The results of a German survey indicate that the prevalence of probable depression was 33.3% and it was 29.6% for probable anxiety in a sample of trans individuals [57]. In the study of Jones and colleagues [58], which also used the HADS to assess depressive and anxious symptomatology, similar results were reported, particularly concerning depressive symptoms. The high values found in our study, especially regarding anxious symptoms, are not surprising and are in line with the literature. Bouman and colleagues [12] indicate that transgender people (particularly trans males) have higher levels of anxiety symptoms suggestive of possible anxiety disorders compared to the general population. In fact, transgender people are disproportionately burdened by poor mental health issues, compared to cisgender people [19].

However, it was expected that participants in the current study would demonstrate lower levels of anxiety and depression, given the high levels of PA previously mentioned and the established relationship between these three variables [39]. This could be explained by the framework of Hendricks’ adapted Minority Stress Model [59]. This model posits that transgender individuals face unique stressors and challenges related to their gender identity. Factors such as discrimination, stigma, gender dysphoria, and lack of social support, which most cisgender individuals do not face, can all contribute to elevated levels of anxiety and depression among transgender individuals, regardless of their level of PA. While PA has beneficial effects on mental health in the cisgender sample, it may not fully mitigate the impact of these stressors in the transgender population. Thus, this could also explain the lack of significant correlations between PA and anxiety and depression in our study.

### 4.3. Body Image

Regarding the values of satisfaction with body image, the results of our study demonstrated reduced values in all the components (face, upper trunk, lower trunk, legs and glutes, body appearance), resulting in a low global SBI. These findings are in line with several studies, which showed that body image concerns were significantly higher in transgender participants compared to cisgender participants [60,61]. In our study, it was also reported that the upper trunk and lower trunk were the body areas where participants expressed the least satisfaction with their body image. This implies that transgender individuals may experience dissatisfaction with various components of their bodies, but certain areas are significant sources of distress, namely the chest or breast area, hips, and overall body shape. This has been observed in the study of van de Grift and colleagues [62], where the hip region and chest size were noted as components of body image with lower satisfaction levels. Similarly, Pulice-Farrow and colleagues [63] and Witcomb and colleagues [15] also found the greatest body dissatisfaction in the chest and in gender-identifying body parts, particularly among trans males.

The low levels of satisfaction with body image observed in transgender individuals can be explained by the Tripartite Influence Model [64], which identifies three primary influences shaping body image perception: media messages, interpersonal influences, and internalization of societal appearance ideals. In fact, traditional and digital media often perpetuate idealized beauty standards that do not represent the diversity of bodies and gender identities within the transgender community. This lack of positive and inclusive representation can lead to feelings of inadequacy and dissatisfaction with body image. Furthermore, interpersonal influences play a critical role, as negative comments and judgments to conform to conventional gender norms can have a significant impact on self-image and contribute to body dissatisfaction among transgender individuals. Additionally, the internalization of social appearance ideals that conflict with a person’s gender identity can lead to internal struggles and decreased self-esteem. Pressure to conform to cisgender beauty norms may further exacerbate feelings of body image dissatisfaction in transgender individuals.

Some studies also suggest that hormone therapy may alleviate body image dissatisfaction and gender dysphoria in certain transgender adults, potentially facilitating the cultivation of a positive body image [65,66]. In this study, although 72% of the participants were currently undergoing transition, the average age at the onset of transition was 21 years, with a mean participant age of 23 years. This demographic profile suggests that a substantial portion of the sample is situated in the early phases of transition. Consequently, the relatively low levels of satisfaction reported may be attributed to the fact that many individuals have yet to experience significant and visible bodily changes associated with hormone therapy, such as changes in fat distribution, muscle mass, and secondary sexual characteristics. In this sense, addressing body dissatisfaction in the interventions for people with gender dysphoria is crucial for transgender individuals in managing their body image satisfaction.

### 4.4. Relationship between Anxiety, Depression, and Satisfaction with Body Image

The study revealed significant correlations between satisfaction with body image and levels of anxiety and depression, with both anxiety and depression showing negative associations with body image satisfaction. This is not surprising, given that body image is a multidimensional construct that involves thoughts, feelings, and perceptions about the body [67] and many transgender individuals experience gender dysphoria, which can trigger a cascade of negative psychological processes [68]. Additionally, the fact that the sample is in the early stages of the transition process seems to enhance the clarity and significance of the relationship between mental health outcomes and body image satisfaction. In fact, an incongruent physical appearance may result in more difficult psychological adaptation and in more exposure to discrimination and stigmatization [62].

Similar results have also been found in other studies, which show that dissatisfaction with body image can lead to feelings of shame and depression, especially when trans individuals face pressure to conform to body ideals [14,42,43]. A multiple group analysis indicated that thin-ideal and muscular-ideal internalization were linked to body shame and depression through body monitoring and appearance comparison among trans individuals [43]. This finding suggests a potential explanation for why global satisfaction with body image accounted for 60% of the variance in depressive symptomatology in our study.

These results can also be understood through the lens of Objectification Theory, which proposes that individuals, particularly women, are frequently objectified and judged based on their physical appearance. This objectification can lead to various negative consequences, including body dissatisfaction, self-objectification, and mental health issues [69]. The internalization of societal ideals of beauty and gender norms can contribute to body dissatisfaction and self-objectification in transgender individuals, as they often face judgment based on their physical appearance for not conforming to stereotypical norms of masculinity or femininity and feel pressure to be accepted and legitimized in society. Comiskey’s model [70] highlights how transgender women are subjected to objectifying gazes and judgments based on societal beauty standards and traditional gender norms, which further contribute to body dissatisfaction, self-objectification, and negative psychological outcomes such as anxiety and depression. This understanding, grounded in Objectification Theory, underscores the importance of addressing societal pressures and challenging restrictive beauty standards in mental health interventions, particularly for individuals vulnerable to anxiety and depression.

### 4.5. Limitations and Future Directions

One limitation of the present study is the absence of exploration into diverse profiles within the sample and the lack of control for other areas of mental health among the participants. Future studies should investigate potential differences in body image satisfaction, PA levels, and symptoms of anxiety and depression based on gender identity (e.g., trans men, trans women, non-binary individuals), generational cohorts (e.g., younger, older adults), and stages of gender transition (e.g., pre-transition, transitioning, post-transition). The study may not control all the potential confounding variables that could influence the relationship between PA, anxiety, depression, and body image. Factors such as access to gender-affirming healthcare, social support, and experiences of discrimination could impact the outcomes among transgender individuals. Furthermore, the study had a cross-sectional design, which entails data collection at a singular time point. This approach does not facilitate the establishment of direct causal relationships, so longitudinal/experimental studies should be considered. For example, a longitudinal study could be conducted to evaluate the effects of an exercise program on transgender individuals and to examine changes in the variables over time, exploring potential causal relationships between exercise and health outcomes. Additionally, future research should analyze the moderating effect of satisfaction with body image, along with the use of more robust methods for monitoring physical activity (e.g., accelerometers) and include a larger sample and involve the analysis of individuals at different stages of transition. By examining individuals across various stages of their transition process, researchers can identify any variations, which facilitates a better understanding of the mechanisms underlying the relationships between PA, satisfaction with body image, and mental health outcomes among transgender individuals.

## 5. Practical Implications

The results of this study provide a comprehensive and insightful understanding of the relationship between the transgender population and various health issues, while also illuminating the complex interplay between PA, satisfaction with body image, and mental health in transgender individuals. The insights from this study hold particular significance within the context of Portugal’s evolving socio-cultural and legal landscape regarding transgender rights and healthcare. By enhancing our understanding of these relationships, researchers, health professionals, and Portuguese policymakers can develop more effective strategies to support the mental health and well-being of transgender individuals.

Reevaluating existing policies and implementing new ones to ensure equitable access to gender-affirming healthcare services is essential within the Portuguese context. Efforts should target reducing discrimination in healthcare settings and investing in education and training for healthcare providers on transgender issues and identities. The professionals should consider assessing and addressing both mental health and body image concerns as part of comprehensive care. Providing inclusive and accessible healthcare that address body dissatisfaction in the interventions for people with gender dysphoria is crucial for transgender individuals in managing their satisfaction with body image and to improve mental health outcomes. Furthermore, the development and implementation of inclusive programs that promote body acceptance and coping strategies, particularly within the context of physical exercise, may also help alleviate the distress related to body image dissatisfaction while also addressing underlying anxiety and depression symptoms.

Specific legislation and policies are needed within the Portuguese context to protect and promote the participation of transgender individuals in physical exercise and sports. This legislation should ensure that gyms and exercise professionals do not focus their evaluation exclusively on the participants’ body image and aesthetics but instead adopt an inclusive approach that acknowledges and celebrates the diversity of bodies while prioritizing the promotion of overall well-being. Creating safe and appropriate spaces, such as inclusive bathrooms and changing facilities, is crucial to fostering transgender individuals’ participation in physical exercise and sports in Portugal. Regular monitoring of variables related to mental health and body image satisfaction by multidisciplinary teams involving psychologists, physicians, and other professionals is also essential.

Having a broad and specific knowledge of the characteristics and concerns of this population is important not only for establishing treatment plans but also for preventing the emergence of mental health disorders, body image disturbances, and complications arising from sedentary behavior or compulsive exercise. The practical implications of the results of this study and the broader literature on physical and psychological health among the transgender population are important and should be considered, especially by Portuguese health professionals, policymakers, and community organizations.

## 6. Conclusions

The results suggest that transgender individuals meet the WHO’s PA recommendations of at least 600 MET minutes per week. However, they exhibit elevated levels of anxiety symptoms and low levels of body image satisfaction, particularly in specific body parts, such as the upper trunk and lower trunk. The results also show that satisfaction with body image was negatively associated with both anxiety and depression symptomatology. Thus, global satisfaction with body image explains 60% of the variance in depressive symptomatology and body appearance explains 46% of the variance in anxiety symptomatology among trans individuals.

## Figures and Tables

**Table 1 healthcare-12-01008-t001:** Descriptive statistics (*n* = 75).

	*n* (%)	Mean		Median (IQR)
		Mean ± SD	(95% CI)	
Age (years)		23.68 ± 6.59	(22.76 to 26.27)	22.50 (6)
SAB				
Female	13 (17.3)			
Male	62 (82.7)			
GI				
Female	62 (82.7)			
Male	13 (17.3)			
APGI (years)		16.09 ± 7.045	(14.33 to 18.20)	15 (6)
AGIE (years)		18.39 ± 7.452	(16.51 to 20.69)	18 (6)
Are you currently in the process of transitioning?				
Yes	54 (72)			
No	21 (28)			
BGT (years)		21.37 ± 6.44	(19.70 to 23.03)	19 (5)
PA (METS)				
Light PA		858 ± 1455.29	(523.17 to 1192.83)	495 (676.50)
Moderate PA		1262.67 ± 2103.23	(778.76 to 1746.58)	560 (1200)
Vigorous PA		1195.73 ± 1879.49	(763.30 to 1628.16)	0 (1680)
TEE		3316.40 ± 3537.58	(2502.48 to 4130.32)	1830 (4638.00)
Mental Health				
Anxious symptomatology		11.67 ± 4.29	(10.68 to 12.65)	11 (6)
Depressive symptomatology		7.41 ± 4.19	(6.45 to 8.38)	8 (6)
Satisfaction with Body Image				
SBI—Face		3.51 ± 0.85	(3.32 to 3.71)	3.43 (1.43)
SBI—Upper Trunk		2.51 ± 1.06	(2.27 to 2.75)	2.33 (1.67)
SBI—Lower Trunk		2.18 ± 1.08	(1.93 to 2.43)	2 (1.75)
SBI—Legs and Glutes		2.60 ± 1.22	(2.31 to 2.88)	2.67 (2)
SBI—Body Appearance		2.71 ± 0.97	(2.48 to 2.93)	2.83 (1.33)
SBI Global		2.82 ± 0.81	(2.63 to 3.01)	2.74 (1.22)

Notes: SAB = sex assigned at birth; GI = gender identity; APGI = age of perception of GI; AGIE = age of GI expression; BGT = beginning age of gender transition; PA = physical activity; TEE = total energy expenditure (METS); SBI = satisfaction with body image.

**Table 2 healthcare-12-01008-t002:** Bivariate correlations across study variables.

Variables	1	2	3	4	5	6	7	8	9	10	11
1. Light PA	1										
2. Moderate PA	0.174	1									
3. Vigorous PA	0.018	0.161	1								
4. TEE	0.525 **	0.752 **	0.634 **	1							
5. Anxiety symptomatology	−0.036	−0.115	−0.120	−0.148	1						
6. Depression symptomatology	−0.061	−0.164	−0.010	−0.128	0.613 **	1					
7. SBI—Face	−0.128	−0.053	0.036	0.065	−0.242 **	−0.515 **	1				
8. SBI—Upper Trunk	−0.097	0.132	−0.003	0.037	−0.351 **	−0.347 **	0.442 **	1			
9. SBI—Lower Trunk	−0.085	0.041	0.231 *	0.112	−0.400 **	−0.490 **	0.514 **	0.613 **	1		
10. SBI—Legs and Glutes	−0.037	0.068	−0.021	0.014	−0.346 **	−0.411 **	0.520 **	0.677 **	0.665 **	1	
11. SBI—Body Appearance	−0.109	0.116	0.054	0.053	−0.460 **	−0.584 **	0.567 **	0.502 **	0.653 **	0.614 **	1

Notes: PA = physical activity; TEE = total energy expenditure (METS); SBI = satisfaction with body image; * *p* < 0.05; ** *p* < 0.01.

**Table 3 healthcare-12-01008-t003:** Multiple regression analysis for anxiety symptomatology.

	R	R^2^	Adj.R^2^	ΔR^2^	F	df1	df2	*p*	Durbin–Watson
**Model 1**	0.49 *	0.24	0.19	0.24	4.39	5	69	0.002	2.43
	** *β* **	** *t* **	** *p* **	**VIF**	**Tolerance**				
SBI—Face	−0.22	−1.63	0.03	1.62	0.62				
SBI—Upper Trunk	−0.32	−1.88	0.02	2.03	0.49				
SBI—Lower Trunk	−0.39	−1.82	0.01	2.35	0.43				
SBI—Legs and Glutes	−0.31	−1.59	0.01	2.49	0.40				
SBI—Body Appearance	−0.44	−2.36	<0.001	2.12	0.47				

**Note.** SBI = satisfaction with body image; R^2^ = r-square; Adj R^2^ = adjusted r-square; ΔR^2^ = differences in r^2^; df = degrees of freedom; *β* = standardized coefficients; *t* = *t* test; R2 = adjusted r-square—explained variance; Δ = differences; F = changes in significance; df = degrees of freedom; *p* = significance value; * *p* < 0.05.

**Table 4 healthcare-12-01008-t004:** Multiple regression analysis for depressive symptomatology.

	R	R^2^	Adj.R^2^	ΔR^2^	F	df1	df2	*p*	Durbin–Watson
**Model 2**	0.63 *	0.40	0.36	0.40	9.23	5	69	<0.001	2.28
	** *β* **	** *t* **	** *p* **	**VIF**	**Tolerance**				
SBI—Face	−0.52	−2.14	<0.001	1.62	0.62				
SBI—Upper Trunk	−0.31	−1.63	0.01	2.03	0.49				
SBI—Lower Trunk	−0.46	−1.84	<0.001	2.35	0.43				
SBI—Legs and Glutes	−0.39	−1.64	0.01	2.49	0.40				
SBI—Body Appearance	−0.58	−2.75	<0.001	2.12	0.47				

**Note.** SBI = satisfaction with body image; R^2^ = r-square; Adj R^2^ = adjusted r-square; ΔR^2^ = differences in r^2^; df = degrees of freedom; *β* = standardized coefficients; *t* = *t* test; R2 = adjusted r-square—explained variance; Δ = differences; F = changes in significance; df = degrees of freedom; *p* = significance value; * *p* < 0.05.

## Data Availability

All data supporting were included on this paper.

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
