# Peer review of "Physical Activity, Anxiety, Depression, and Body Image in Trans Individuals: An Exploratory Study"

_healthcare, 2024, doi:10.3390/healthcare12101008_

Round 1
Reviewer 1 Report
Comments and Suggestions for Authors
1. Abstract: abbreviation were not deciphered when the were introduced for the first time.
2. The authors stated that "Satisfaction with body image was negatively associated with anxiety (r = 0, 441, p < 0.01) and depression symptomatology (r = 0, 600, p < 0.01).", however, these are positive correlations as r-values are positive. This is an error which should be fixed.
3. Lines 80-101: Long paragraphs are unwanted. Please divide this paragraph into 2 paragraphs (line 92 as separator).
4. Internal consistency reliability should be used (not just internal consistency).
5. Small italicized n should be used (instead of N) to indicate sample size of non-population based sample.
6. Why did you not present correlations between SBI subscales and other variables in Table 2? This should be presented.
7. It seems somewhat incomprehensible to present univariate regression analysis in Tables 3 and 4. Multivariate regression analysis with multiple predictors would be beneficial in order to reveal specific psychological targets for anxiety and depression symptoms among trangender people. Please recalculate. Otherwise, the current results seem to repeat your correlational analysis when you calculate correlation between SBI subscales and anxiety and depression symptoms.
8. "Overall, the results suggest that transgender individuals meet the WHO's PA recommendations.". this sentence seems confusing. Please proofread your paper carefully.
Author Response
Reviewer 1
- Abstract: abbreviation were not deciphered when the were introduced for the first time.
Response: All abbreviations have been deciphered and clarified in the manuscript. Thank you.
- The authors stated that "Satisfaction with body image was negatively associated with anxiety (r = 0, 441, p < 0.01) and depression symptomatology (r = 0, 600, p < 0.01).", however, these are positive correlations as r-values are positive. This is an error which should be fixed.
Response: We apologize for the error in stating the correlations. The correct statement should indicate that satisfaction with body image was negatively associated with anxiety (r = -0.441, p < 0.01) and depression symptomatology (r = -0.600, p < 0.01). This correction has been made in the manuscript. We appreciate your careful review.
- Lines 80-101: Long paragraphs are unwanted. Please divide this paragraph into 2 paragraphs (line 92 as separator).
Response: We have divided the paragraph into two shorter paragraphs. Thank you for your comment.
- Internal consistency reliability should be used (not just internal consistency).
Response: We thank the reviewer for their comment. There are a several manners to calculate internal consistency. In present study we included cronback alpha considering the total sample size (n=75). Considering the total sample it is not possible to calculate composite reliability or McDonald's ωt. For that it is necessary to conduct a Confirmatory Factor Analysis for determine the factorial weights of each item on respective factor as recommended by several authors (e.g., Byrne, 2016; Hair et al., 2019).
- Small italicized nshould be used (instead of N) to indicate sample size of non-population based sample.
Response: It has been corrected.
- Why did you not present correlations between SBI subscales and other variables in Table 2? This should be presented.
Response: We thank the reviewer for their comment. It was a typo. Sorry. The bivariate correlation across SBI subscales were included.
- It seems somewhat incomprehensible to present univariate regression analysis in Tables 3 and 4. Multivariate regression analysis with multiple predictors would be beneficial in order to reveal specific psychological targets for anxiety and depression symptoms among trangender people. Please recalculate. Otherwise, the current results seem to repeat your correlational analysis when you calculate correlation between SBI subscales and anxiety and depression symptoms.
Response: We appreciate the reviewer's comment. Indeed, a Multivariate regression would make perfect sense if the theoretical rationale predicted it. That is, the multiple predictors are the subscales of the SBI, and as such, they are all correlated with each other, so none can assume the role of independent and another dependent, as demonstrated by the validation of the instrument:
- Rodrigues, F., Monteiro, D., Flores, P., & Forte, P. (2021). On Redefining the Body Image Satisfaction Questionnaire: A Preliminary Test of Multidimensionality. HealthCare 9 (7), 876. https://doi.org/10.3390/healthcare9070876
Therefore, our choice was to perform a simple linear regression considering the subscales of the SBI as predictors and symptoms of anxiety and depression as the dependent variable.
- "Overall, the results suggest that transgender individuals meet the WHO's PA recommendations.". this sentence seems confusing. Please proofread your paper carefully.
Response: We have reviewed the sentence and made the necessary revisions for clarity. Thank you.
Reviewer 2 Report
Comments and Suggestions for Authors
The manuscript titled "Physical Activity, Anxiety, Depression and Body Image in Trans Individuals: An Exploratory Study" explores critical aspects of the physical and mental health of transgender individuals, particularly focusing on their engagement in physical activity (PA), levels of anxiety and depression, and satisfaction with body image. The study is timely and addresses a gap in the literature regarding the characterization of transgender individuals.
However, several areas require improvement and clarification to enhance the manuscript's scholarly contribution and relevance. Specifically: In the Introduction, the manuscript should emphasize the diversity within transgender experiences, highlighting that transgender identities encompass a spectrum beyond binary notions of gender. It should also acknowledge non-binary identities explicitly. Some references could be included to support this discussion.
In addition, the assertion regarding PA levels among transgender individuals should reflect the influence of legal and socio-cultural contexts. Variations in PA levels may exist across different regions or countries due to varying levels of inclusiveness and acceptance. On this topic you can see:
- Monaco, S., & Corbisiero, F. (2022). Urban sexuality across Europe. Do LGBT neighborhoods matter?. Polish Sociological Review, 219(3), 351-366;
- Singh, B. (2022). Understanding legal frameworks concerning transgender healthcare in the age of dynamism. Electronic Journal of Social and Strategic Studies, 3, 56-65.
- Stroumsa, D. (2014). The state of transgender health care: policy, law, and medical frameworks. American journal of public health, 104(3), e31-e38.
In the section devoted to the "Current Study", authors should specify the geographical context of the study (Portugal) and provide relevant information regarding the inclusiveness and legal norms pertinent to transgender individuals in Portugal. This contextualization is essential for understanding the study findings within a specific socio-cultural framework.
As for the analysis, authors should improve this part using an intersectional lens, considering diverse profiles within the sample. Exploring potential differences based on gender identity, generational cohorts, and stages of transition could provide valuable insights.
Finally, "practical implications" should be rooted in the Portuguese context, emphasizing the relevance of the findings within the socio-cultural and legal landscape of Portugal. This would enhance the applicability of the findings for stakeholders in Portugal.
I also have some formal recommendations:
- Avoiding beginning paragraphs with "Indeed..." or "Overall" to maintain a formal and concise writing style.
- Please, consider presenting practical implications in a dedicated paragraph rather than as a sub-section within the discussion.
Author Response
Reviewer 2
The manuscript titled "Physical Activity, Anxiety, Depression and Body Image in Trans Individuals: An Exploratory Study" explores critical aspects of the physical and mental health of transgender individuals, particularly focusing on their engagement in physical activity (PA), levels of anxiety and depression, and satisfaction with body image. The study is timely and addresses a gap in the literature regarding the characterization of transgender individuals.
Response: Thank you very much for your comments. We will try to respond to all comments and suggestions which we believe will improve the quality of the manuscript.
However, several areas require improvement and clarification to enhance the manuscript's scholarly contribution and relevance. Specifically: In the Introduction, the manuscript should emphasize the diversity within transgender experiences, highlighting that transgender identities encompass a spectrum beyond binary notions of gender. It should also acknowledge non-binary identities explicitly. Some references could be included to support this discussion.
Response: We appreciate the reviewer's comment. We have made revisions to address the aspects you highlighted for improvement in the Introduction section. We have also included relevant references.
In addition, the assertion regarding PA levels among transgender individuals should reflect the influence of legal and socio-cultural contexts. Variations in PA levels may exist across different regions or countries due to varying levels of inclusiveness and acceptance. On this topic you can see:
- Monaco, S., & Corbisiero, F. (2022). Urban sexuality across Europe. Do LGBT neighborhoods matter?. Polish Sociological Review, 219(3), 351-366;
- Singh, B. (2022). Understanding legal frameworks concerning transgender healthcare in the age of dynamism. Electronic Journal of Social and Strategic Studies, 3, 56-65.
- Stroumsa, D. (2014). The state of transgender health care: policy, law, and medical frameworks. American journal of public health, 104(3), e31-e38.
Response: Thank you for your suggestion. We have incorporated this aspect into the revised manuscript.
In the section devoted to the "Current Study", authors should specify the geographical context of the study (Portugal) and provide relevant information regarding the inclusiveness and legal norms pertinent to transgender individuals in Portugal. This contextualization is essential for understanding the study findings within a specific socio-cultural framework.
Response: We thank the reviewer for their comment. The geographical context, along with relevant information about the inclusiveness and legal norms pertaining to transgender individuals in Portugal was included in revised manuscript.
As for the analysis, authors should improve this part using an intersectional lens, considering diverse profiles within the sample. Exploring potential differences based on gender identity, generational cohorts, and stages of transition could provide valuable insights.
Response: We thank the reviewer for their interesting comment. However, we do not collect this kind of information. We included this question on limitations section and future avenues.
Finally, "practical implications" should be rooted in the Portuguese context, emphasizing the relevance of the findings within the socio-cultural and legal landscape of Portugal. This would enhance the applicability of the findings for stakeholders in Portugal.
Response: Thank you for your comment. We have revised and reformulated the practical implications to align with the specific socio-cultural and legal context of Portugal.
I also have some formal recommendations:
- Avoiding beginning paragraphs with "Indeed..." or "Overall" to maintain a formal and concise writing style.
Response: Thank you for the recommendation. All paragraphs that began with "Indeed" and "Overall" have been revised accordingly.
- Please, consider presenting practical implications in a dedicated paragraph rather than as a sub-section within the discussion.
Response: The practical implications were presented in a dedicated paragraph in the revised manuscript. Thank you for your recommendation.
Reviewer 3 Report
Comments and Suggestions for Authors
The submitted manuscript entitled “Physical Activity, Anxiety, Depression and Body Image in Trans Individuals: An Exploratory Study” addressed an important issue which is mental and physical health of people with body dysphoria. The Authors recruited 75 individuals at various levels of transition process. Although the overall quality of the study is relatively high, the following issues should be regarded to improve the manuscript.
#1. p.1: “The term transgender or trans refers to a person whose gender identity differs from 29 the sex that was assigned to them at birth”. Please, refer to the newest approaches to this phenomenon as it is treated in the ICD-11 (e.g., discuss the changes in the nomenclature around this phenomenon).
#2. The Authors could describe the law regulations regarding process of transition. In various countries, the law regulations are different and could influence the context of the transition process.
#3. Please, describe in details which were the regression models summarized in the Table 3-4. It seems that the models consisted of one predictor. If my understanding is correct, why all predictors were not included in one model.
#4. Please, indicate which were the definitions and calculations behind moderate-intensity PA, vigorous-intensity PA, and light-intensity PA.
#5. While discussion the results, please do not only refer to the results of the previous studies, but also to some theories which may suggest the reasons of the associations (e.g., between body dissatisfaction in dysphoria and PA).
#6. Suggestions based on the results should be revised. The Authors did not examine the role of gender-affirmation as an intervention. We only knew from the study that higher body dissatisfaction is associated with more depressiveness. Thus, the suggestions should be for example, to address body dissatisfaction in the interventions for people with gender dysphoria.
#7. Another limitation could be a lack of control of other areas of mental health of the participants.
#8. Please indicate how the sample size was determined. Which are the limitations of the sample size.
Comments on the Quality of English LanguageThe manuscript needs edition for typos.
Author Response
Reviewer 3
The submitted manuscript entitled “Physical Activity, Anxiety, Depression and Body Image in Trans Individuals: An Exploratory Study” addressed an important issue which is mental and physical health of people with body dysphoria. The Authors recruited 75 individuals at various levels of transition process. Although the overall quality of the study is relatively high, the following issues should be regarded to improve the manuscript.
Response: Thank you for your feedback. We will consider all comments and suggestions that we think will enhance the quality of our manuscript and aim to respond accordingly.
#1. p.1: “The term transgender or trans refers to a person whose gender identity differs from 29 the sex that was assigned to them at birth”. Please, refer to the newest approaches to this phenomenon as it is treated in the ICD-11 (e.g., discuss the changes in the nomenclature around this phenomenon).
Response: We appreciate the reviewer's comment. The newest approaches have been referenced in the revised manuscript.
#2. The Authors could describe the law regulations regarding process of transition. In various countries, the law regulations are different and could influence the context of the transition process.
Response: The legal regulations pertaining to the transition process have been incorporated into the revised manuscript. Thank you for your recommendation.
#3. Please, describe in details which were the regression models summarized in the Table 3-4. It seems that the models consisted of one predictor. If my understanding is correct, why all predictors were not included in one model.
Response: We appreciate the reviewer's comment. Two regression models were performed (one for anxiety symptoms and another for depressive symptoms). The independent variables are dimensions of the Body Image Satisfaction Questionnaire. As such, they do not represent a single predictor, but rather several (i.e., SBI – Face; SBI – Upper Trunk; SBI – Lower Trunk; SBI – Legs and Glutes; SBI – Body Appearance; SBI Global).
#4. Please, indicate which were the definitions and calculations behind moderate-intensity PA, vigorous-intensity PA, and light-intensity PA.
Response: The definitions and calculations for moderate-intensity PA, vigorous-intensity PA, and light-intensity PA, as outlined by Craig et al. (2003), are based on the concept of Metabolic Equivalent Tasks (METs) and specific activity levels. Moderate-intensity PA refers to activities that result in a moderate increase in heart rate and breathing, with a threshold defined at 4.0 METs. Vigorous-intensity PA encompasses activities that lead to a substantial increase in heart rate, breathing, sweating, or notably harder breathing than usual, such as running, aerobics, or fast cycling, with a threshold defined at 8.0 METs. Light-intensity PA includes activities causing slight increases in heart rate and breathing, like walking, with a threshold set at 3.3 METs.
To calculate MET minutes a week, multiply the MET value given (light activity = 3.3, moderate activity = 4, vigorous activity = 8) by the minutes the activity was carried out and again by the number of days that that activity was undertaken.
#5. While discussion the results, please do not only refer to the results of the previous studies, but also to some theories which may suggest the reasons of the associations (e.g., between body dissatisfaction in dysphoria and PA).
Response: We thank the reviewer for their comment. We have included relevant theories to complement our discussion.
#6. Suggestions based on the results should be revised. The Authors did not examine the role of gender-affirmation as an intervention. We only knew from the study that higher body dissatisfaction is associated with more depressiveness. Thus, the suggestions should be for example, to address body dissatisfaction in the interventions for people with gender dysphoria.
Response: We appreciate the reviewer's comment. The suggestions have been revised, and they now include the specific recommendation to address body dissatisfaction in interventions for individuals with gender dysphoria.
#7. Another limitation could be a lack of control of other areas of mental health of the participants.
Response: We appreciate the reviewer's suggestion. We have incorporated this limitation into the revised manuscript.
#8. Please indicate how the sample size was determined. Which are the limitations of the sample size.
Response: We thank the reviewer for their comment. The sample size was calculated using G*Power 3.1.9.2 (Faul et al., 2017), based on the following input parameters: effect size (f² = 0.3), α = 0.05, and statistical power = 0.95. The minimum required sample size was determined to be 55 subjects, which was adhered to in the current study. This information has been included in revised manuscript.
Reviewer 4 Report
Comments and Suggestions for Authors
The topic of the study is interesting and the effort made by the authors is appreciated. However, there are a series of basic problems at work, as I explain below.
-Introduction: Some aspects that are introduced are not fully explained sufficiently for the reader to understand them. For example, how does body image satisfaction moderate the relationship between physical activity and mental health? Next, it is indicated that, in the same way as physical activity, body image satisfaction also seems to be associated with mental health, but it is not clarified what this moderating effect initially talked about consists of.
Nor is it very clear why a reference to the relationship between satisfaction with body image and physical health has been included so briefly, if reference is then made again to mental health.
-Participants: The sample size for a descriptive study is not representative, nor has a random selection procedure been applied that would later allow the generalization of the results to the population.
Data Analysis and Results: If the aim is to deepen the relationships between the variables analyzed as predictors of mental health (anxiety and depression), as it appears in the objectives of the study, in the summary and in the discussion, the data analysis strategy, with regard to the type of regression models built, does not seem the most appropriate. From the results of the Pearson correlations, the effect sizes (R2) can be obtained, without the need to build all those independent simple regression models, of which the unbiased estimator of the effect size is not reported either (adjusted R2). According to the initial approach, and with the data available, the joint influence of all the variables proposed as independent should have been analyzed, as well as the interaction effects between them, as mentioned in different parts of the manuscript.To do this, multiple regression models should have been built, in which physical activity would have also been included as a predictor, since it does not seem to be sufficiently justified to only analyze the contribution of each dimension of body satisfaction, as well as the variable overall, in anxiety and depression, leaving aside physical activity, despite its importance having been justified from the beginning of the manuscript. An analysis of the possible interaction effects between the variables should also have been carried out, based on regression analysis or factorial ANOVA, for example.
-Discussion: explanations are proposed to understand why the participants, although they present a high level of physical activity, also show high levels of anxiety and other possible reasons are referred to, although it could have been verified if it has to do with the possible moderating effect that could affect body image satisfaction and, although it is introduced at the beginning of the study, it has not been analyzed in the work.
Although in Discussion section it is stated that the interaction between the variables has been analyzed, the truth is that the work only presents a basic descriptive analysis and a bivariate analysis. Through this type of data analysis, interaction effects between the variables or their joint influence cannot be obtained or inferred.
Author Response
Reviewer 4
The topic of the study is interesting and the effort made by the authors is appreciated. However, there are a series of basic problems at work, as I explain below.
Response: Thank you very much for your feedback. We have tried to respond to all comments and improve the manuscript according to the suggestions and comments made.
-Introduction: Some aspects that are introduced are not fully explained sufficiently for the reader to understand them. For example, how does body image satisfaction moderate the relationship between physical activity and mental health? Next, it is indicated that, in the same way as physical activity, body image satisfaction also seems to be associated with mental health, but it is not clarified what this moderating effect initially talked about consists of.
Response: We appreciate the reviewer's comment. Body image has been identified as a moderator in the relationship between physical activity and mental health (particularly anxiety), indicating that the impact of physical activity on anxiety levels depends on perceived body image. Specifically, individuals with a heightened awareness of body image tend to exhibit a stronger association between physical activity and anxiety (Han et al., 2023). We have clarified the nature of this moderating effect in the revised manuscript.
Nor is it very clear why a reference to the relationship between satisfaction with body image and physical health has been included so briefly, if reference is then made again to mental health.
Response: We thank the reviewer for their comment. It was an typo. This reference has been removed.
-Participants: The sample size for a descriptive study is not representative, nor has a random selection procedure been applied that would later allow the generalization of the results to the population.
Response: We thank the reviewer for their comment. The calculation of power sample was included in revised manuscript.
Data Analysis and Results: If the aim is to deepen the relationships between the variables analyzed as predictors of mental health (anxiety and depression), as it appears in the objectives of the study, in the summary and in the discussion, the data analysis strategy, with regard to the type of regression models built, does not seem the most appropriate. From the results of the Pearson correlations, the effect sizes (R2) can be obtained, without the need to build all those independent simple regression models, of which the unbiased estimator of the effect size is not reported either (adjusted R2). According to the initial approach, and with the data available, the joint influence of all the variables proposed as independent should have been analyzed, as well as the interaction effects between them, as mentioned in different parts of the manuscript.To do this, multiple regression models should have been built, in which physical activity would have also been included as a predictor, since it does not seem to be sufficiently justified to only analyze the contribution of each dimension of body satisfaction, as well as the variable overall, in anxiety and depression, leaving aside physical activity, despite its importance having been justified from the beginning of the manuscript. An analysis of the possible interaction effects between the variables should also have been carried out, based on regression analysis or factorial ANOVA, for example.
Response: We thank the reviewer for their comment. The R2 is reported on the tables. Please see the footnote. Indeed, a Multivariate regression would make perfect sense if the theoretical rationale predicted it. That is, the multiple predictors are the subscales of the SBI, and as such, they are all correlated with each other, so none can assume the role of independent and another dependent, as demonstrated by the validation of the instrument:
- Rodrigues, F., Monteiro, D., Flores, P., & Forte, P. (2021). On Redefining the Body Image Satisfaction Questionnaire: A Preliminary Test of Multidimensionality. HealthCare 9 (7), 876. https://doi.org/10.3390/healthcare9070876
Therefore, our choice was to perform a simple linear regression considering the subscales of the SBI as predictors and symptoms of anxiety and depression as the dependent variable.
Regarding physical activity, it was not included in the regression analysis since it did not show a significant correlation with the other variables, thus failing to meet an essential prerequisite for conducting a regression analysis (Ho, 2014). It is not possible to conduct a ANOVA, since the variables under analysis are not group variables.
-Discussion: explanations are proposed to understand why the participants, although they present a high level of physical activity, also show high levels of anxiety and other possible reasons are referred to, although it could have been verified if it has to do with the possible moderating effect that could affect body image satisfaction and, although it is introduced at the beginning of the study, it has not been analyzed in the work.
Response: We thank the reviewer for their comment. In the discussion, we aim to propose potential explanations for these findings based on existing research and literature. In addition to the factors we discussed, the use of the International Physical Activity Questionnaire (IPAQ) for monitoring physical activity may have contributed to explaining these data. For this study, it was not our objective to analyze the moderating effect of body image satisfaction, although this could be considered in future research, along with the use of more robust methods for monitoring physical activity (e.g., accelerometers).
Although in Discussion section it is stated that the interaction between the variables has been analyzed, the truth is that the work only presents a basic descriptive analysis and a bivariate analysis. Through this type of data analysis, interaction effects between the variables or their joint influence cannot be obtained or inferred.
Response: We appreciate the reviewer's comment. The transgender population continues to be underrepresented in scientific research, leading to a lack of studies that thoroughly explore specific variables and provide a comprehensive characterization, as demonstrated by Oliveira et al. (2022). The primary objective of our study was to conduct a comprehensive assessment of physical activity levels, anxious and depressive symptomatology, and satisfaction with body image, while also examining the relationships among these variables. Therefore, our choice was to perform a simple linear regression considering the subscales of the SBI as predictors and symptoms of anxiety and depression as the dependent variable.
Round 2
Reviewer 1 Report
Comments and Suggestions for Authors
Regarding my previous comment 4 (see below), I just meant that the term "internal consistency" is incomplete. A complete term is "internal consistency reliability". Adding the world "reliability" would be beneficial.
My previous comment 4 and your response to it are presented below.
(4. Internal consistency reliability should be used (not just internal consistency).
Response: We thank the reviewer for their comment. There are a several manners to calculate internal consistency. In present study we included cronback alpha considering the total sample size (n=75). Considering the total sample it is not possible to calculate composite reliability or McDonald's ωt. For that it is necessary to conduct a Confirmatory Factor Analysis for determine the factorial weights of each item on respective factor as recommended by several authors (e.g., Byrne, 2016; Hair et al., 2019).).
------------------------------------------------------------------------
Regarding my previous comment 7 (see below), I meant that all five SBI subscales could be put into the two regression models as predictors. So, the first regression model predicts anxiety symptoms and the second one predicts depression symptopathology. In each of the models, five SBI subscales are considered as predictors. This type of analysis could provide more specific information, especially in indicating which SBI subscales associated with symptopathology more strongly, and which ones less strongly or even not associated when controlling for the common variance. The paper by Rodrigues et al. (2021) indicates that SBI subscales represent specific components of the BIS construct, which should explain the specific variance of predictive ability. So, I encourage the authors to use more comprehensive regression analysis with multiple predictors in order to understand which SBI components are stronger associated with psychopathology. This should help in evaluating more specific and relevant psychotherapeutical targets. Otherwise, your univariate regression analysis would represent just simple repeatings of the univariate correlations.
Note: A SBI Global score should not be included as a predictor in the above described models, because the regression models will not converge.
My previous comment 7 and your response to it are presented below.
(7. It seems somewhat incomprehensible to present univariate regression analysis in Tables 3 and 4. Multivariate regression analysis with multiple predictors would be beneficial in order to reveal specific psychological targets for anxiety and depression symptoms among trangender people. Please recalculate. Otherwise, the current results seem to repeat your correlational analysis when you calculate correlation between SBI subscales and anxiety and depression symptoms.
Response: We appreciate the reviewer's comment. Indeed, a Multivariate regression would make perfect sense if the theoretical rationale predicted it. That is, the multiple predictors are the subscales of the SBI, and as such, they are all correlated with each other, so none can assume the role of independent and another dependent, as demonstrated by the validation of the instrument:
• Rodrigues, F., Monteiro, D., Flores, P., & Forte, P. (2021). On Redefining the Body Image Satisfaction Questionnaire: A Preliminary Test of Multidimensionality. HealthCare 9 (7), 876. https://doi.org/10.3390/healthcare9070876
Therefore, our choice was to perform a simple linear regression considering the subscales of the SBI as predictors and symptoms of anxiety and depression as the dependent variable.)
------------------------------------------------------------------
Please use zeroes before full stops in numbers. Please correct some inconsistencies (e.g., line 188 etc.).
--------------------------------------------------------------------
Lines 148-149: Please eliminate the last sentence ("This information...").
--------------------------------------
In general, the authors should specify for what statistical analyses sample size was calculated.
------------------------------------------------
Author Response
Regarding my previous comment 4 (see below), I just meant that the term "internal consistency" is incomplete. A complete term is "internal consistency reliability". Adding the world "reliability" would be beneficial.
My previous comment 4 and your response to it are presented below.
(4. Internal consistency reliability should be used (not just internal consistency).
Response: We thank the reviewer for their comment. There are a several manners to calculate internal consistency. In present study we included cronback alpha considering the total sample size (n=75). Considering the total sample it is not possible to calculate composite reliability or McDonald's ωt. For that it is necessary to conduct a Confirmatory Factor Analysis for determine the factorial weights of each item on respective factor as recommended by several authors (e.g., Byrne, 2016; Hair et al., 2019).).
R: We thank the reviewer for their comment. We would like to ask apolozige for some misinterpretation. Therefore the word realibility has been included.
Regarding my previous comment 7 (see below), I meant that all five SBI subscales could be put into the two regression models as predictors. So, the first regression model predicts anxiety symptoms and the second one predicts depression symptopathology. In each of the models, five SBI subscales are considered as predictors. This type of analysis could provide more specific information, especially in indicating which SBI subscales associated with symptopathology more strongly, and which ones less strongly or even not associated when controlling for the common variance. The paper by Rodrigues et al. (2021) indicates that SBI subscales represent specific components of the BIS construct, which should explain the specific variance of predictive ability. So, I encourage the authors to use more comprehensive regression analysis with multiple predictors in order to understand which SBI components are stronger associated with psychopathology. This should help in evaluating more specific and relevant psychotherapeutical targets. Otherwise, your univariate regression analysis would represent just simple repeatings of the univariate correlations.
Note: A SBI Global score should not be included as a predictor in the above described models, because the regression models will not converge.
My previous comment 7 and your response to it are presented below.
(7. It seems somewhat incomprehensible to present univariate regression analysis in Tables 3 and 4. Multivariate regression analysis with multiple predictors would be beneficial in order to reveal specific psychological targets for anxiety and depression symptoms among trangender people. Please recalculate. Otherwise, the current results seem to repeat your correlational analysis when you calculate correlation between SBI subscales and anxiety and depression symptoms.
Response: We appreciate the reviewer's comment. Indeed, a Multivariate regression would make perfect sense if the theoretical rationale predicted it. That is, the multiple predictors are the subscales of the SBI, and as such, they are all correlated with each other, so none can assume the role of independent and another dependent, as demonstrated by the validation of the instrument:
- Rodrigues, F., Monteiro, D., Flores, P., & Forte, P. (2021). On Redefining the Body Image Satisfaction Questionnaire: A Preliminary Test of Multidimensionality. HealthCare 9 (7), 876. https://doi.org/10.3390/healthcare9070876
Therefore, our choice was to perform a simple linear regression considering the subscales of the SBI as predictors and symptoms of anxiety and depression as the dependent variable.)
R: We thank the reviewer for their comment. We ask to apologize for some misinterpretation. In this regard, a new table has been added as suggested by the reviewer. In addition, we take this opportunity to correct a mistake in terms of signal direction (i.e., all regression coefficients are negative and not positive). Sorry. The SBI-Global has been removed.
Please use zeroes before full stops in numbers. Please correct some inconsistencies (e.g., line 188 etc.).
R: It has been corrected. Thanks.
R: Removed. Thanks.
In general, the authors should specify for what statistical analyses sample size was calculated.
R: The power of sample size was determined for the main analysis (i.e., regression). It was included in the revised manuscript.
Reviewer 3 Report
Comments and Suggestions for Authors
The Authors addressed all my suggestions. I believe that the revised version of the manuscript could be suggested for publication.
Comments on the Quality of English LanguageThe language needs some minor stylistic editions.
Author Response
The Authors addressed all my suggestions. I believe that the revised version of the manuscript could be suggested for publication.
The language needs some minor stylistic editions.
R: Thank you very much for the comment. Some linguistic and grammatical changes have been made throughout the document.
Reviewer 4 Report
Comments and Suggestions for Authors
It is appreciated that the authors have taken the comments into account to apply all the improvements. With them the manuscript has increased its quality.
Author Response
It is appreciated that the authors have taken the comments into account to apply all the improvements. With them the manuscript has increased its quality.
R: Thank you very much for the comment.